# The Effect of Different Ferrule Configurations and Preparation Designs on the Fatigue Performance of Endodontically Treated Maxillary Central Incisors: A 3D Finite Element Analysis

Mehmet Gökberkkaan Demirel [1,*] and Reza Mohammadi [2]

¹ Department of Prosthodontics, Faculty of Dentistry, Necmettin Erbakan University, 42090 Konya, Turkey
² Faculty of Dentistry, Necmettin Erbakan University, 42090 Konya, Turkey;
reza.mohammadi@ogr.erbakan.edu.tr
* Correspondence: kaandemirel@erbakan.edu.tr; Tel.: +90-534-040-5100

**Abstract:** The presence of an adequate circumferential ferrule is of great importance for the prosthetic rehabilitation of endodontically treated teeth. However, there may not be an adequate circumferential ferrule effect. This study aimed to evaluate the fatigue performance of endocrown and post–core restorations applied to different configurations without an adequate ferrule effect using the finite element method and endeavors to offer a comprehensive perspective on the challenging rehabilitation of maxillary incisors with excessive coronal substance loss, addressing a notable gap in research and providing valuable insights for clinicians. The goal is to address this gap in research and contribute valuable insights that could be beneficial for practitioners. The maxillary central incisor was selected, and lithium disilicate (LS) and a polymer-infiltrated ceramic network (PICN) were used in post–core groups with no ferrule effect (PC0); a 2 mm ferrule effect on the buccal wall (PC1); a 2 mm ferrule effect on the buccal-mesial wall (PC2); a 2 mm ferrule effect on the buccal–mesial–palatal wall (PC3); and a 2 mm circumferential ferrule effect (PC4) In the endocrown groups, an external retention group with a circumferential ferrule (ECER) and an inner retention form group (ECIR) were prepared. Fatigue performance was examined by applying a 150 N oblique load. The evaluation of the fatigue performance of the restorative materials shows that LS always had more successful fatigue performance results, while the post–core groups were generally more successful in terms of dentin survival. In situations where there is insufficient circumferential ferrule, the application of endocrowns is likely to result in a less successful prognosis for survival.

**Keywords:** digital dentistry; endocrown; fatigue performance; ferrule; finite element analysis; 3D analysis for accuracy





## 1. Introduction

Endodontically treated teeth are generally considered weaker, and fractures are more likely than in vital teeth [1]. While the main difference in biomechanical behavior is associated with [2] extensive coronal tissue loss, it has been reported in the literature that the reduced elasticity of dentin tissue and the reduced moisture content in the tissue may be due to other causes [3].

The method used to restore teeth with extensive coronal tissue loss after endodontic treatment is generally a full crown applied with post and core [4]. However, it is considered necessary to find a suitable ferrule effect to reduce the risk of tooth fracture, prolong the survival of the restoration, and optimize the distribution of stress in the surrounding tissues [1,5,6]. The ferrule effect describes the circumferential dentin walls, which are vertical in relation to the margins of restoration [1] and should ideally be 2–3 mm high [2]. However, it may not always be clinically possible to attain a circumferential ferrule at a uniform height. In addition, crown elevation procedures carry surgical risks, can damage the supporting tooth tissue or bone, and increase the time and cost involved in treatment [7].

Given these circumstances, the clinician may have to apply treatment with many different ferrule configurations. In particular, configurations without labial and palatal walls are more challenging for the clinician. Zhang et al. and Pantaleon et al. reported the importance of the presence of the palatal wall [1,8], Figueiredo et al. argued that different ferrule configurations did not affect biomechanical behavior [9], and Ding et al. stated that the internal retention shape would be useful in terms of root strength [5]. However, the effect of different ferrule configurations on rehabilitation with post–core systems has not been sufficiently studied.

In dentistry, endocrowns, which were introduced around three decades ago with the development of adhesive technologies and new materials, are widely clinically used today. Endocrowns are prostheses prepared as monoblocks and cemented to the pulp chamber's internal wall and margins. They adequately restore the tooth's anatomy and prevent bacterial recolonization by closing the root canal, thus ensuring longer survival after endodontic treatment [10]. Clinical trials have not reached a consensus on the biomechanical behaviors of endocrown and post–core restorations. While Forberger and Göhring [11] and Silva-Sousa et al. [12] argue that post–core restorations are preferable, Biacchi [11], Basting [13], and Xixi et al. [14] report that endocrown restorations exhibit greater fracture strength.

Glass ceramics have brittle behavior and low tensile strength, which can lead to catastrophic fractures even at low stresses. With advances in dental ceramic formulations, glassy phase infiltrated lithium disilicate (LS) ceramic systems have been introduced [15]. These ceramics are used in fully anatomical monolithic crowns, particularly for maxillary incisional tooth restoration due to their increased durability and preferred transparency. Although the long-term success rate of glass ceramics is satisfactory, the development of polymer-infiltrated ceramic network (PICN) ceramics has become increasingly popular in dentistry due to its clinically successful use [16]. It has also been suggested that polymer-based composites combine the properties of both the ceramic and polymer, resulting in hybrid materials with performance and mechanical behavior closer to that of teeth [17].

This study aimed to evaluate the fatigue performance of post–core and endocrown restorations (made with adhesively cemented LS and PICN restorative materials) applied to different ferrule configurations by means of finite element analysis. Clinicians may think that the fatigue performance of post–core and endocrown restorations is affected by different materials and ferrule configurations (H1), and a null hypothesis (H0) was proposed in that neither the restorative materials and restoration types nor different ferrule configurations would affect fatigue performance.

## 2. Materials and Methods

### 2.1. Preparation of Samples

The solid model was prepared by placing a maxillary central tooth in a silicone mold simulating the maxillary arch and obtaining a model with an elastomeric impression material. The incisal and middle thirds of the crown were reduced with a round end cylinder burr, preserved a 2 mm coronal tooth structure from the cemento-enamel junction, and a 1 mm margin was prepared based on the chamfer design. Root canal treatment was applied to the tooth, and after one day, the gutta-percha was removed, leaving a 5 mm apical seal. The post was selected to be the thickest, not exceeding 1/3 of the root diameter, and placed in the canal.

For this study, two different ceramic materials, lithium disilicate (LS) and a polymer-infiltrated ceramic network (PICN), were used, and seven different configurations were designed (Figure 1). A tomography image was taken for the circumferential ferrule endocrown group (ECER) prior to post application. After application of the post, the post was cut to the appropriate height for the PC4 group, and the core was prepared by connecting the resin composite to the ferrule. In the PC3 group, the distal wall of the circumferential ferrule was reduced, and the missing portion was restored with the resin composite and incorporated into the core. In the PC2 group, the palatal wall was reduced, and the missing portion was restored with the resin composite and incorporated into the core. In the PC1

group, the mesial wall was reduced, and the missing portion was restored with the resin composite and incorporated into the core. In the PC0 group, the buccal wall was elevated, and the missing portion was restored with the resin composite and incorporated into the core. Finally, for the ECIR group, the post was removed, and the root canal was refilled with gutta-percha. After each stage, a scan was performed with a cone-beam computed tomography device (Morita 3D Accuitomo 170 (J Morita Mfg. Corp., Kyoto, Japan)). The dimensions of the imaging volume were diameter 40 mm × height 40 mm in a cylindrical form in the X-ray conversion center. The images were taken under standard parameters of 90 kVp (X-ray tube voltage) and 5 mA (the value of the electric current). The images were taken using the parameters 160 sqm and 17.5 s exposure time. The isotropic (Table 1) and orthotropic (Table 2) properties of the dental tissues and materials used in finite element analysis have been presented in Tables 1 and 2, respectively.

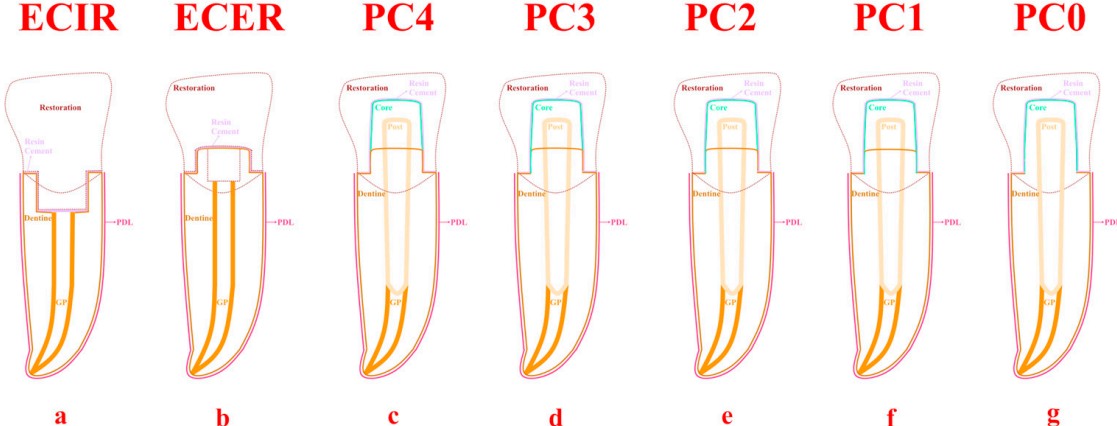

**Figure 1.** Scheme of samples; ECIR—(**a**): no ferrule, endocrown; ECER—(**b**): 2 mm circumferential ferrule, endocrown; PC4—(**c**): 2 mm circumferential ferrule, post–core; PC3—(**d**): 2 mm mesial, palatal, and distal ferrule, post–core; PC2—(**e**): 2 mm mesial and distal ferrule, post–core; PC1—(**f**): 2 mm distal ferrule, post–core; PC0—(**g**): no ferrule, post–core.

**Table 1.** Mechanical properties of tooth tissues and dental ceramics.

| Material | Young's Modulus (MPa) | Poisson's Ratio | Compressive Strength (MPa) | Flexural Strength (MPa) | Shear Strength (MPa) | Fracture Toughness (MPa m$^{1/2}$) | Microhardness (H$_V$) | Reference |
|---|---|---|---|---|---|---|---|---|
| LS | 102,700 | 0.22 | - | 356.7 | - | 2.8 | 676.7 | |
| PICN | 30,100 | 0.23 | - | 135.8 | - | 1.4 | 261.7 | |
| Dentin | 18,600 | 0.31 | 297 | 105.5 | 12–138 | | | |
| Cement | 7700 | 0.3 | 262 | 98 | 40 | | | |
| Dental resin composite | 12,000 | 0.30 | | | | | | [18] |
| Pulp | 2 | 0.45 | | | | | | |
| PDL | 68,900 | 0.45 | | | | | | |
| Gutta-percha | 140,000 | 0.45 | | | | | | |
| Cortical bone | 13,700 | 0.30 | | | | | | |
| Spongy (cancellous) bone | 1370 | 0.3 | | | | | | |

**Table 2.** The orthotropic properties of fiber-reinforced composite resin posts.

| Material | Young's Modulus (MPa) | Poisson's Ratio | Shear Modulus (MPa) | Tensile Failure Limit (MPa) | Compressive Failure Limit (MPa) | Reference |
|---|---|---|---|---|---|---|
| Glass fiber-reinforced post | X, 37.0<br>Y, 9.5<br>Z, 9.5 | XY, 0.27<br>XZ, 0.34<br>YZ, 0.27 | XY, 3.1<br>XZ, 3.5<br>YZ, 3.1 | 180.1–215.8 | 118.8–151.5 | [19] |

MPa: megapascals, X,Y,Z: axis in the Cartesian coordinate system.

## 2.2. Numerical Simulation

The results were obtained as Digital Imaging and Communications in Medicine (DICOM) files. The files obtained were transferred to the Mimics (Materialise Interactive Medical Image Control System (Mimics 20.00, Leuven, Belgium)) software, and different masks were created for each tooth tissue (dentin and pulp) and other parts (post and core). Once the three-dimensional objects of each mask had been created, they were converted into Standard Tessellation Language (STL) files. The resulting STL files were transferred to the design software (Dental Cad. 3.1 Rijeka, EXOCAD, Darmstadt, Germany), and the STL files were extracted after designing the coronal restoration for each model with the same data. After transferring the resulting STL files to a reverse engineering program (Geomagic Design X 2020/03), a periodontal ligament (PDL) (0.25 mm thick) and cortical bone (0.5 mm thick) were created and extracted as Standard for the Exchange of Product Model Data (STEP) files. The generated elements were transferred to a finite element analysis program (ABAQUS 2020, Dassault Systems Simulation Corp., Johnston, RT, USA) for fatigue performance, and the mesh convergence was determined based on the report by Gonder et al. [20]. Afterward, 150 N force (on the palatal incisal edge at a 45° angle) was applied to each group for 1000 cycles. The data were obtained as maximum principal stress values (Pmax).

## 2.3. Calculation of Fatigue Performance

The fatigue life evaluation criterion of the restoration material and dentin tissue compares the maximum principal stress generated during loading with the stress-life (S-N) diagram. The S-N curve is a function of the number of cycles (N) to failure and represents the stress amplitude ($\sigma_a$). The fatigue values for the crown material were determined by the three-point bending tests as in previous studies. The fatigue behavior of the materials was calculated using the non-linear Basquin's Equation (1) developed by Mutluay et al. [21].

$$\sigma = A(N)^B \tag{1}$$

The A and B values for the crown material are demonstrated in Table 3. Wöhler curves for enamel and dentin were taken from the literature and plotted for precise inverse loading ($\sigma_m = 0$). Mathematically, the Equation (2) to be applied for any other mean stress ($\sigma_m \neq 0$) has been demonstrated in previous studies [22,23].

$$\sigma_a = (\sigma_f - \sigma_m)\,(2N)^b \tag{2}$$

**Table 3.** Coefficient and exponent constants of the fatigue curves of dental ceramics and dentin.

|  | A (MPa) | B | $\sigma_f$ (MPa) | b | References |
|---|---|---|---|---|---|
| LS | 95.845 | −0.012 |  |  | [19] |
| PICN | 61.40 | −0.043 |  |  | [19] |
| Dentin |  |  | 247 | −0.111 | [23] |

The constant values for dentin and enamel, $\sigma_f$ and b, are shown in Table 3.

### 3. Results

As a result of this study, Pmax values were found for the prosthetic restoration (Figure 2), resin cement (Figure 3), resin composite core (Figure 4), fiber post (Figure 5), dentin (Figure 6), and periodontal ligament (Figure 7).

The highest of the Pmax values found in the restoration was found in the ECIR group for both types of restoration, while the lowest value was found in the ECER group. The Pmax values resulting from restoration increased in the post–core groups as the amount of ferrule decreased (Figure 2).

The Pmax values obtained in resin cement showed an inversely proportional result with ferrule volume similar to the restoration in the post–core groups, whereas the endocrown groups showed relatively good results. However, the PC4 group was identified as the most successful group. While the highest Pmax value occurred in the ECER group, the Pmax values seen in the dentin tissue increased as the ferrule volume decreased in the post–core groups (Figure 3).

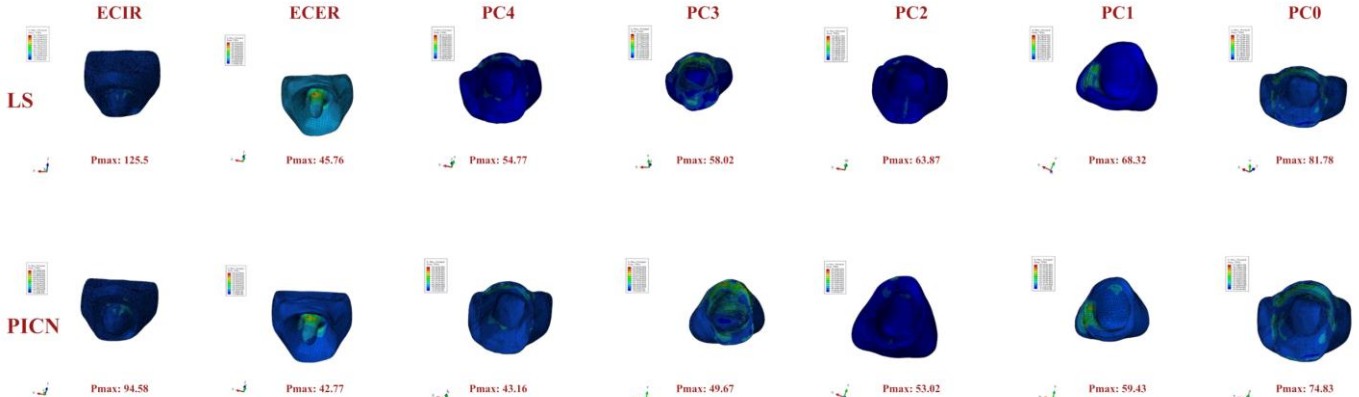

**Figure 2.** The Pmax values observed in the restoration; ECIR: no ferrule, endocrown; ECER: 2 mm circumferential ferrule, endocrown; PC4; 2 mm circumferential ferrule, post–core; PC3: 2 mm mesial, palatal, and distal ferrule, post–core; PC2: 2 mm mesial and distal ferrule, post–core; PC1: 2 mm distal ferrule, post–core; PC0: no ferrule, post–core; Pmax: maximum principal stress values; LS: lithium disilicate; PICN: polymer-infiltrated ceramic network.

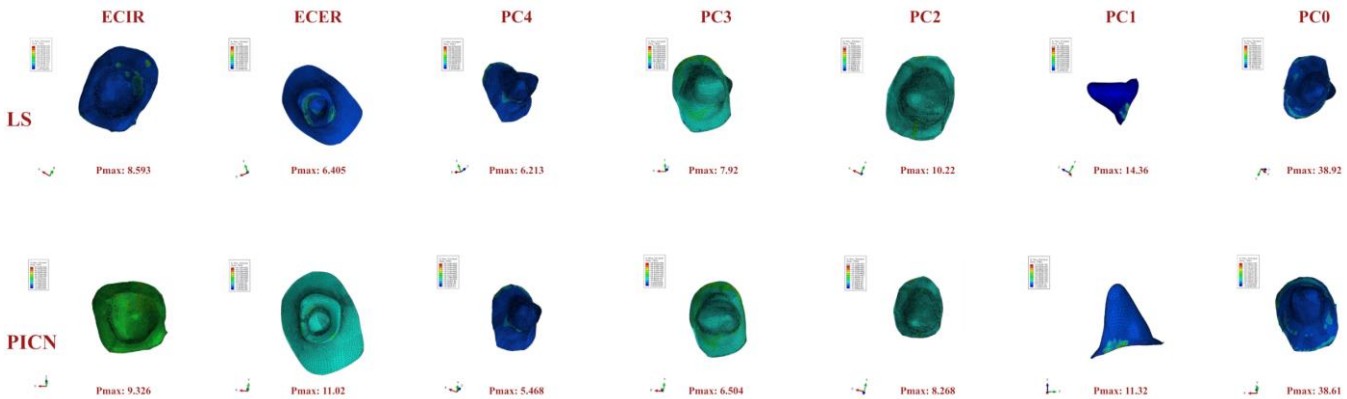

**Figure 3.** The Pmax values observed in the resin cement; ECIR: no ferrule, endocrown; ECER: 2 mm circumferential ferrule, endocrown; PC4: 2 mm circumferential ferrule, post–core; PC3: 2 mm mesial, palatal, and distal ferrule, post–core; PC2: 2 mm mesial and distal ferrule, post–core; PC1: 2 mm distal ferrule, post–core; PC0: no ferrule, post–core; Pmax: maximum principal stress values; LS: lithium disilicate; PICN: polymer-infilrated ceramic network.

The Pmax values in the resin composite core and fiber post revealed a completely inversely proportional result. For the resin composite core, the Pmax values increased as the ferrule volume increased (Figure 4), whereas for the fiber post, the Pmax values

decreased as the ferrule volume decreased (Figure 5). For both, the Pmax values in the PICN groups were higher than in the LS groups.

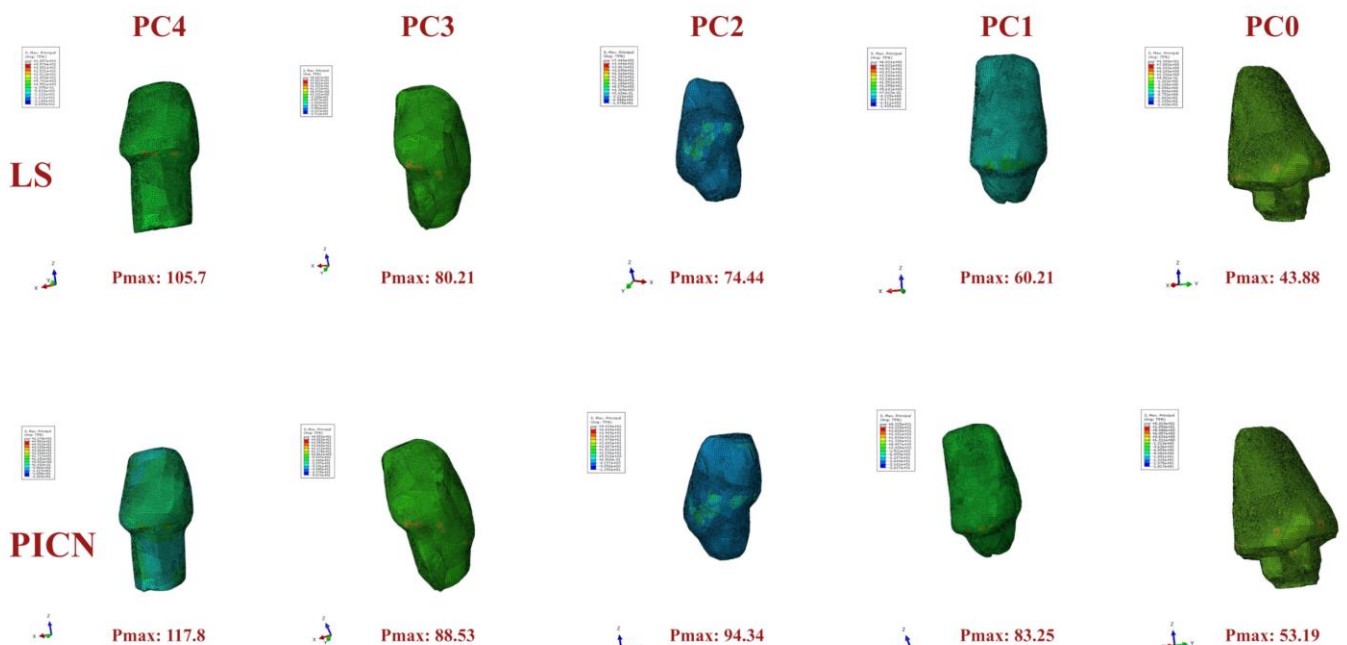

**Figure 4.** The Pmax values observed in the resin composite core; ECIR: no ferrule, endocrown; ECER: 2 mm circumferential ferrule, endocrown; PC4: 2 mm circumferential ferrule, post–core; PC3: 2 mm mesial, palatal, and distal ferrule, post–core; PC2: 2 mm mesial and distal ferrule, post–core; PC1: 2 mm distal ferrule, post–core; PC0: no ferrule, post–core; Pmax: maximum principal stress values; LS: lithium disilicate; PICN: polymer-infiltrated ceramic network.

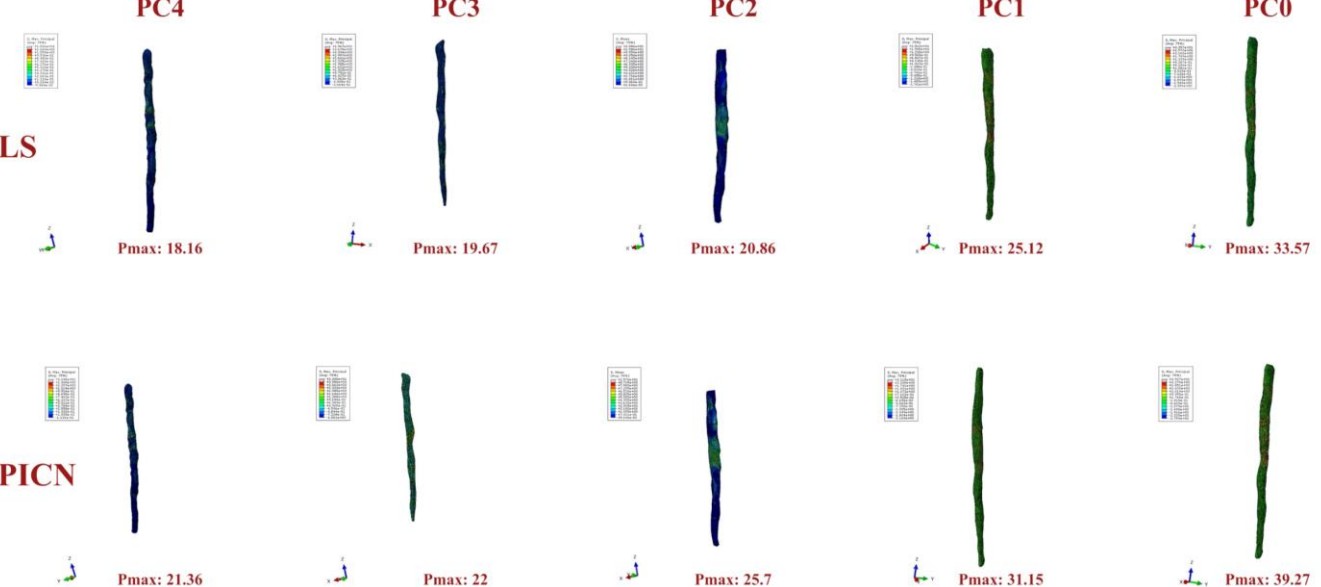

**Figure 5.** The Pmax values observed in the post; PC4: 2 mm circumferential ferrule, post–core; PC3: 2 mm mesial, palatal, and distal ferrule, post–core; PC2: 2 mm mesial and distal ferrule, post–core; PC1: 2 mm distal ferrule, post–core; PC0: no ferrule, post–core; Pmax: maximum principal stress values; LS: lithium disilicate; PICN: polymer-infiltrated ceramic network.

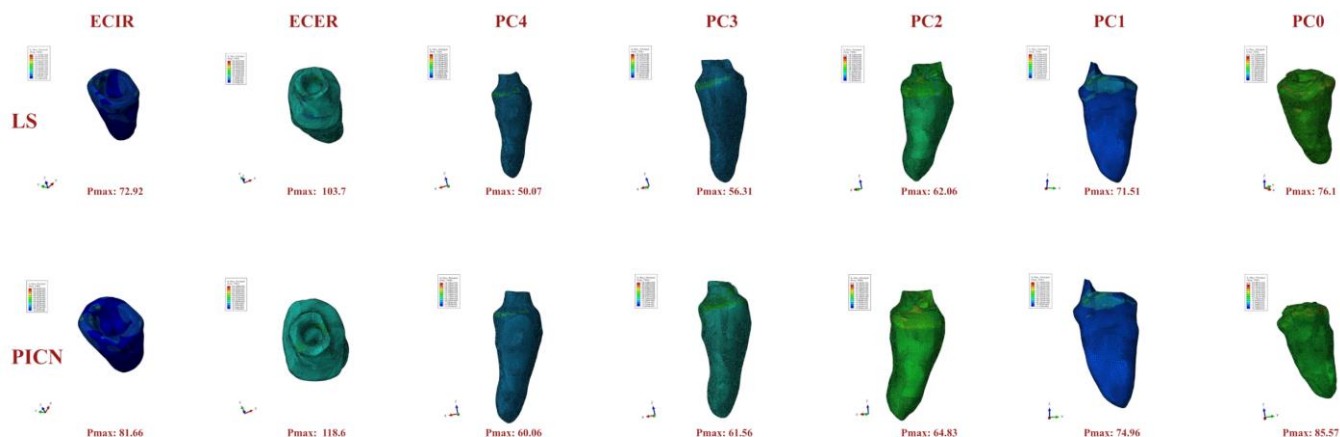

**Figure 6.** The Pmax values observed in the dentin tissue; ECIR: no ferrule, endocrown; ECER: 2 mm circumferential ferrule, endocrown; PC4: 2 mm circumferential ferrule, post–core; PC3: 2 mm mesial, palatal, and distal ferrule, post–core; PC2: 2 mm mesial and distal ferrule, post–core; PC1: 2 mm distal ferrule, post–core; PC0: no ferrule, post–core; Pmax: maximum principal stress values; LS: lithium disilicate; PICN: polymer-infiltrated ceramic network.

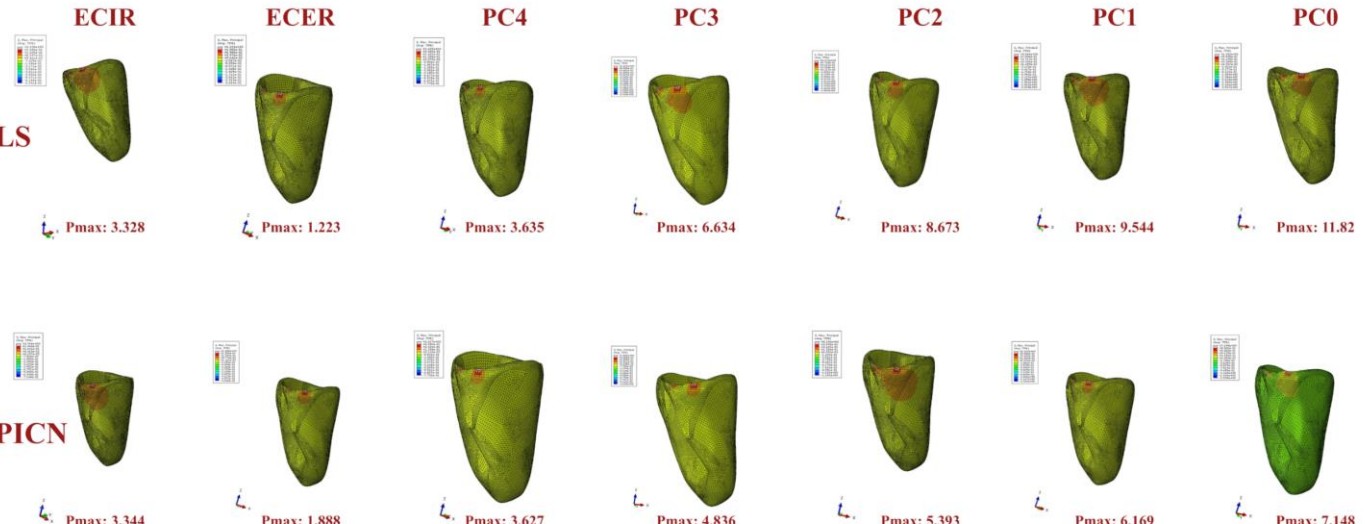

**Figure 7.** The Pmax values observed in the periodontal ligament; ECIR: no ferrule, endocrown; ECER: 2 mm circumferential ferrule, endocrown; PC4: 2 mm circumferential ferrule, post–core; PC3: 2 mm mesial, palatal, and distal ferrule, post–core; PC2: 2 mm mesial and distal ferrule, post–core; PC1: 2 mm distal ferrule, post–core; PC0: no ferrule, post–core; Pmax: maximum principal stress values; LS: lithium disilicate; PICN: polymer-infiltrated ceramic network.

In dentine loads, the post–core groups showed significantly better results than the endocrown groups. Even the PC1 group, which was ferruled in only one wall, demonstrated a better result than the endocrown groups. The Pmax values in the PICN groups were higher than in the LS groups (Figure 6).

In contrast to dentin, endocrowns showed less stress on the PDL. Again, as the ferrule volume decreased in the post–core groups, the Pmax values increased dramatically. The best result was observed in the ECER group (Figure 7).

When the Pmax values found in the restoration were examined, LS exhibited values greater than the PICN for each group (Figure 2). For the Pmax values observed in dentin (Figure 6), the fiber post (Figure 5), and the resin composite cores (Figure 4), the PICN revealed higher values in each group.

In terms of the Pmax values found in the resin cement (Figure 3), the Pmax values were higher in the PICN for the endocrown groups, which led to an opposite result in the post–core groups, resulting in higher values for LS. For the PDL, this was identified as being the exact opposite of resin cement.

The fatigue performance was estimated by applying the Pmax values of restorations to Equation (1) and the Pmax values of dentin to Equation (2) (Table 2). When the fatigue performances of restoration and dentin were compared, LS demonstrated better results each time.

The dentin tissue was found to be less durable than the restorative material in the estimated fatigue performance values calculated for the restorative material (Figure 8) and the dentin tissue (Figure 9). Of the materials, configurations prepared with LS showed significantly better fatigue performance than those prepared with the PICN. It can be said that the survival of dentin tissue will be longer in all configurations prepared with LS. ECER showed the most unsuccessful fatigue performance in relation to dentin, while the post–core groups generally showed successful results. The ECIR group with internal retention showed a result close to PC1, while the post–core group prepared with two or more walls showed better fatigue performance. The PC4 group with a circumferential ferrule demonstrated the best fatigue performance.

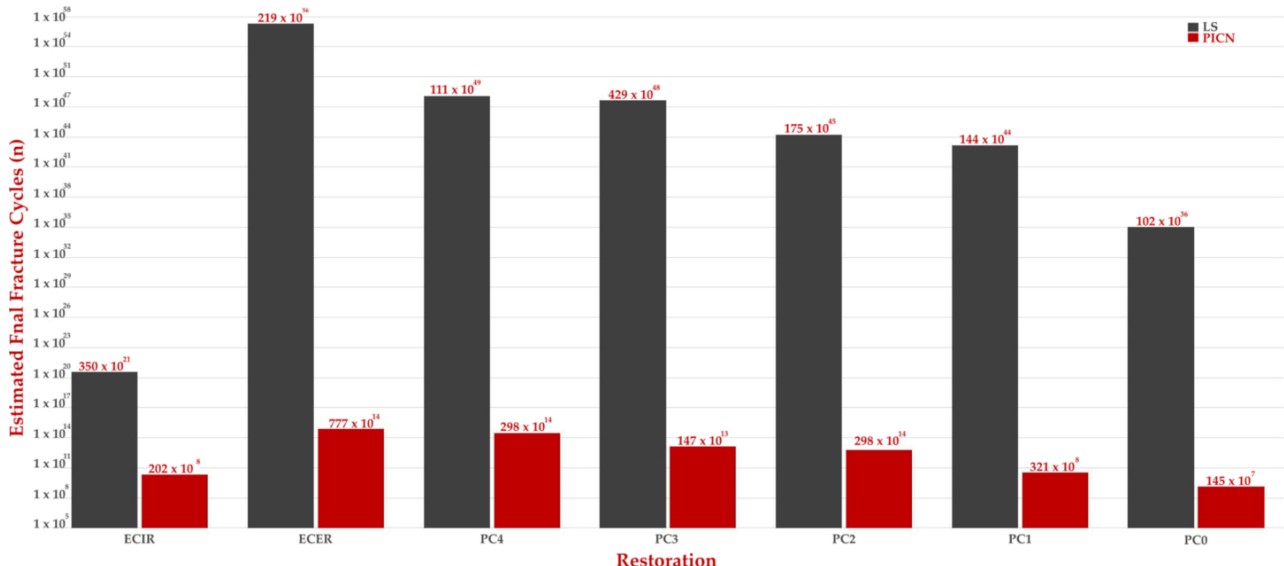

**Figure 8.** The estimated fatigue performance cycles for the restoration; ECIR: no ferrule, endocrown; ECER: 2 mm circumferential ferrule, endocrown; PC4: 2 mm circumferential ferrule, post–core; PC3: 2 mm mesial, palatal, and distal ferrule, post–core; PC2: 2 mm mesial and distal ferrule, post–core; PC1: 2 mm distal ferrule, post–core; PC0: no ferrule, post–core; Pmax: maximum principal stress values; LS: lithium disilicate; PICN: polymer-infiltrated ceramic network.

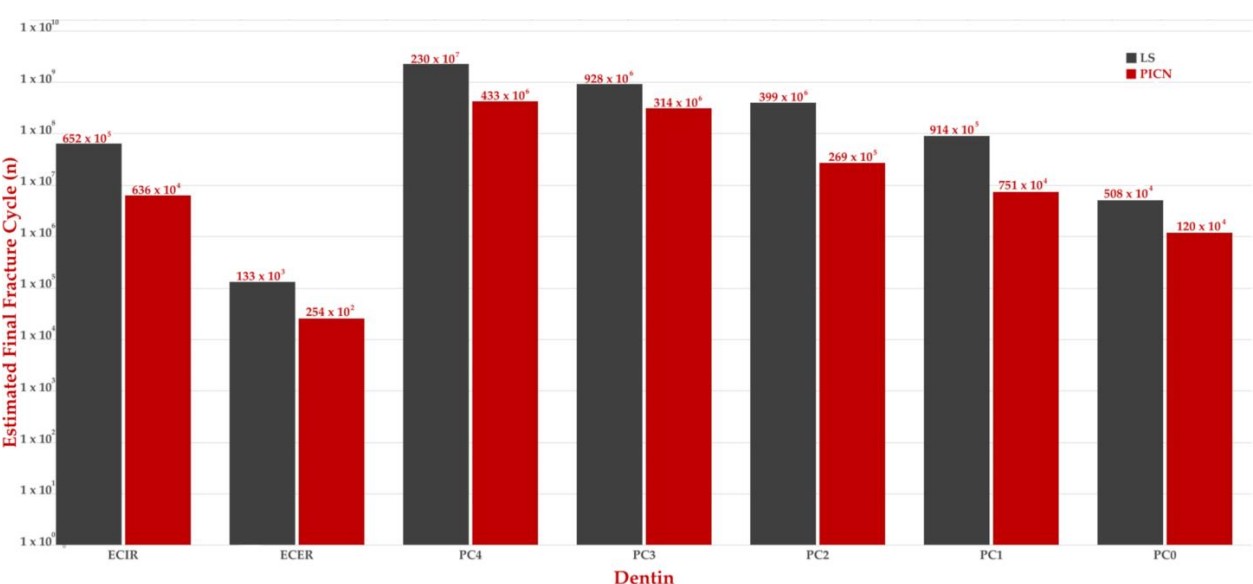

**Figure 9.** The estimated fatigue performance cycles for the dentin; ECIR: no ferrule, endocrown; ECER: 2 mm circumferential ferrule, endocrown; PC4: 2 mm circumferential ferrule, post–core; PC3: 2 mm mesial, palatal, and distal ferrule, post–core; PC2: 2 mm mesial and distal ferrule, post–core; PC1: 2 mm distal ferrule, post–core; PC0: no ferrule, post–core; Pmax: maximum principal stress values; LS: lithium disilicate; PICN: polymer-infiltrated ceramic network.

## 4. Discussion

This study aimed to evaluate the fatigue performance of post–core and endocrown restoration types applied to different ferrule configurations by means of finite element analysis. As a result of the FEM, all of the configurations studied were found to affect the fatigue performance. As such, the initial hypothesis was rejected.

This study demonstrated the effect of residual coronal structure on fatigue performance in the rehabilitation of maxillary central incisors following endodontic treatment. These data are supported by other studies [8,24–26] and confirm the favorable effect of the ferrule effect on mechanical strength and prognosis. The existence of the ferrule effect is expected to increase resistance to lever loads and positively affect the structure by distributing stresses more uniformly [27]. As can be seen in Figure 1, the ECIR and PC0 groups, which had no ferrule effect, had the greatest loads on the restoration materials, while the most successful group was PC4, where all four walls were intact. However, the presence of even one coronal wall (PC1) is critical for fatigue performance (Figure 2), as shown in the study by Kawasaki et al. [28]. These situations are also supported by estimated fatigue performance cycles (Table 2). In addition, the results found for resin cement were similar. This may also indicate that in configurations where the ferrule effect cannot be created, decementation will occur faster. However, the Pmax value detected in resin cement for ECIR was relatively low. This may be because the stresses exerted by the internal vertical walls providing internal retention are transferred to the restoration rather than to the resin cement.

Although PC4, PC3, and PC2 showed similar results when looking at post loads, PC1 and PC0 showed a rapid increase in Pmax values. There must therefore be at least two intact walls to ensure an ideal distribution of the load on the post. As the number of intact walls decreased, the incoming loads increased rapidly. This result is consistent with the work of Gupta et al. [26].

The Pmax values in dent in tissue were lower in post–core restorations. These findings also support those of other studies [12,29,30]. Endocrown restorations, a reconstruction option for endodontically treated teeth with extensive coronal damage, are particularly indicated for molars [13,31]. In endocrowns, the elimination of the hybrid post/composite core component reduces the number of interfaces that need to be connected, thereby reducing the negative effects of hybrid layer disruption [13]. Endocrowns are also thicker

in the occlusal section and can also be expected to have higher fracture resistance than conventional crowns [32]. However, these findings apply to molar endocrowns.

To simulate the clinical conditions as closely as possible, the lateral and axial movements of the tooth in the alveolar socket should be simulated [33]. This was attempted by adding the visco-hyperelastic structure of the PDL to the study [34].

When comparing restorative materials, LS showed a higher stress distribution in the restoration material, while the PICN showed higher values in all other groups tested. These results are in accordance with previous studies [35,36]. This means that the LS material absorbs the incoming loads more in its structure while transmitting less to other parts. Reducing load transfer, particularly to the dentin and post, may be important to avoid catastrophic failure.

Maximum principal stresses were greater in LS, which has a higher modulus of elasticity, than in the PICN when examined in terms of restorations and showed a lower risk of fracture in the PICN, which has more elastic properties, than in LS. This observation may mean that endocrowns made with resin matrix ceramics, which have a better stress distribution, will last longer [37,38]. However, this study also showed that the LS material transmitted less stress to the enamel. From this point of view, it can be assumed that the failures that may occur in endocrown restorations made with resin matrix ceramics will also be catastrophic [39], as the Pmax values occurring in both the post and dentin tissues were greater in all groups prepared with the PICN restorative material.

This study investigated the effect on fatigue performance of prosthetic rehabilitation types to be used in cases where different dentin tissue configurations are present in endodontically treated anterior teeth with excessive coronal material loss. There are a small number of academic studies looking at the use of endocrowns in anterior teeth. In their study, Silva-Sousa et al. examined post–core and endocrown restorations with a 2 mm circumferential ferrule, and as a result, they reported that the post–core groups had better fracture resistance [12]. Dejak and Mlotkowski examined circumferentially ferruled post–core and endocrown restorations using FEM and reported that Pmax values in dentin were higher in the endocrown groups. Endocrowns used in the anterior teeth have been biomechanically likened to short posts by many authors. Compared to ideal posts at 2/3 of the root length, short posts are approximately two to five times weaker [30,40–42]. In addition, Forberger and Göhring proved that endocrowns provide a less successful marginal continuity than fiber posts [11]. As the ferrule volume decreased in the post–core groups, the Pmax values increased. As mentioned earlier, the presence of the ferrule effect increases resistance to lever loads and positively affects the structure by distributing stresses more uniformly. In addition, previous studies have shown that the presence of a ferrule strengthens the entire dentin-restorative structure [43,44].

The evaluation of the fatigue performance of the restoration materials shows that the LS material has significantly better fatigue performance than the PICN material. LS materials are preferred for use in endocrown restorations due to their strength and ability to bond to dental tissues [30]. In this study, the circumferential ferrule endocrown group (ECER) showed the highest fatigue performance value. However, lithium disilicate endocrowns are also prone to damage or loosen anterior teeth under physiological loads. In their study, Bankoğlu Güngör et al. found that incisors restored with lithium disilicate ceramic endocrowns had higher fracture resistance than those restored with post and core, but they also reported that they caused more tooth fractures [29]. This is consistent with the results of the study of Silva-Soares et al. [12]. In this study, although the ECER group showed the highest fatigue performance value in terms of the restoration material, it demonstrated a very low fatigue performance value for dentin. This also supports the study by Bankoğlu Güngör et al. [29].

In this study, the fatigue performance of dentin was examined, and post–core configurations with two or more walls showed better results than in the endocrown groups. Sufficient bonding space is critical for the retention and stabilization of endocrowns [45]. The smaller pulp chamber area in the incisors may have reduced the retention forces. In

addition, the diameter of the pulpal extension of the restoration is very small, which may have resulted in poor stress transfer between the crown and the dentin [46]. In post–core systems, the post placed in the root canal allows the root and restoration material to act as one, and the stress distribution is more appropriate as the retention increases [46,47]. These situations may have caused post–core systems to exhibit better fatigue performance for dentin. However, when the ECIR group prepared with an inner retention form was examined for dentin survival, it demonstrated fatigue performance comparable to the PC2 group. Especially in maxillary incisors with an inadequate ferrule in endodontically treated teeth, establishing an inner retention can be a viable option. Ding et al. reported in their study that restorations prepared with an inner retention technique can be a particularly favorable option for teeth with thin-walled dentin [5].

FEM analysis was used to investigate the hypothesis of this study. FEM serves as a computational tool in the analysis of diverse aspects pertaining to dental restorations, biomaterials, restorative techniques, and prosthetic designs, particularly with regard to stress distribution under varying loading conditions [48]. This numerical simulation technique facilitates the thorough examination and quantification of the biomechanical reactions exhibited by intricate dental structures [30]. It is routinely employed to scrutinize stress patterns in dental biomechanical investigations. The modeling process assumed that all parts used for different configurations were perfectly joined and that the properties of the material used were isotropic and linear. Therefore, any errors arising during the production or preparation of materials were ignored. In addition to these limitations, a formula was applied to the 1000 cycle results to calculate the estimated fatigue performance values. It would be appropriate to validate future studies with the support of experimental research and to use higher cycle values.

## 5. Conclusions

This study strives to present a thorough perspective on the intricate rehabilitation of endodontically treated maxillary incisors facing extensive coronal loss by addressing a significant research gap, and it aims to furnish valuable insights for clinicians engaged in clinical practice.

The results obtained within the limitations of this study are as follows:

1.  The presence of a circumferential ferrule is critical in endodontically treated teeth with excessive coronal tissue loss. However, post–core restorations, which are applied in cases where at least two walls are intact, can be expected to be successful.
2.  Endocrowns for anterior teeth are not a good treatment alternative.
3.  If the remaining number of coronal walls is 1 or 0, an endocrown with internal retention may also be a good treatment alternative.
4.  The use of more rigid restorative materials, such as LS, may be recommended for endodontically treated teeth with excessive coronal tissue loss.

**Author Contributions:** Conceptualization, M.G.D.; methodology, M.G.D. and R.M.; software, R.M.; validation, M.G.D. and R.M.; formal analysis, R.M.; investigation, M.G.D. and R.M.; resources, M.G.D.; data curation, M.G.D. and R.M.; writing—original draft preparation, M.G.D.; writing—review and editing, M.G.D.; visualization, M.G.D.; supervision, M.G.D. and R.M.; project administration, M.G.D.; funding acquisition, M.G.D. All authors have read and agreed to the published version of the manuscript.

**Funding:** This research received no external funding.

**Institutional Review Board Statement:** Not applicable.

**Informed Consent Statement:** Not applicable.

**Data Availability Statement:** The raw data supporting the conclusions of this article will be made available by the authors on request.

**Conflicts of Interest:** The authors declare no conflicts of interest.

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
