# Peer review of "The Effect of Different Ferrule Configurations and Preparation Designs on the Fatigue Performance of Endodontically Treated Maxillary Central Incisors: A 3D Finite Element Analysis"

_applsci, doi:10.3390/app14041355_

Round 1
Reviewer 1 Report
Comments and Suggestions for Authors
Dear Authors,
Thank you for submitting this manuscript. The paper is quite interesting because it refers to a significant topic. Overall, this is a good and well-written article. I would like to suggest some points to the Authors:
1. The abstract should include a short statement on the current research gap and question to show why this study is unique and worthy of publication.
2. Please add more references in the Introduction section, lines 58 – 69.
3. Lines 241 – 253, 288 - 295 – there aren’t any references for these statements, please add them.
4. The authors should add the null and working hypotheses of the study and highlight them by adding "H0" and "H1"
5. Please add the limitations of this study.
6. In the Conclusion section, please describe the essential significance of this study.
Thank you in advance for all the corrections. Good luck!
Author Response
Comments 1: The abstract should include a short statement on the current research gap and question to show why this study is unique and worthy of publication. |
Response 1: We express our deepest gratitude for bringing this matter to our attention. The requested revisions have been implemented in lines 16-18. |
Comments 2: Please add more references in the Introduction section, lines 58 – 69. |
Response 2: We sincerely appreciate your insights. We have incorporated the suggested references into the manuscript. |
Comments 3: Lines 241 – 253, 288 - 295 – there aren’t any references for these statements, please add them. Response 3: Your observations are indeed valid, and we sincerely appreciate your insights. We have incorporated the suggested references into the manuscript. However, we would like to clarify that the comments made in lines 277-280 pertain to aspects where there is insufficient information in the existing literature, and we believe our interpretations are suitable for the relevant findings. We express our gratitude for bringing these points to our attention. Comments 4: How to obtain parameters for finite element analysis of complex materials like PICN? |
Response 4: Null and working hypotheses have been added to lines 76-80. We appreciate your valuable suggestions and thank you for bringing this to our attention. |
Comments 5: Please add the limitations of this study. |
Response 5: With all due respect, we would like to bring to your attention that we discussed the limitations of our study in lines 353-361. Should you have any comments or suggestions for these limitations, we would be more than happy to revised them. |
Comments 6: In the Conclusion section, please describe the essential significance of this study |
Response 6: Thank you for pointing this out. We extend our deepest gratitude for bringing this matter to our attention. The requested modifications have been made in lines 349-352. |
[We extend our sincere gratitude for your kind remarks at the beginning and your heartfelt wishes at the end. We appreciate your valuable suggestions, and we hope that we have been able to incorporate the necessary revisions as per your insightful comments. Wishing you well.]

Reviewer 2 Report
Comments and Suggestions for Authors
This article provides a substantial amount of information in a structured form of descriptive text, figures and tables, but with no introduction being denoted as such. (Introduction as a separate chapter ?)
The abstract is also not divided into structural sections. (When not necessary ignore the comment)
However, in order to be understood more clearly, it needs significant language improvement. Throughout the text there are many examples of confused choice of words, grammar mistakes and too complicated syntax, which disrupts the train of thought and readability. (whole Results chapter!)
In the text, the results are often assessed as "good“, "better“, "successful“, or "unsuccessful“ without being referred to the range that defines such classification. (no statistical significance to support statements (numbers)! in rows 23-25, whole Results Chapter! and rows 311-319)
Therefore, I recommend this article to be reconsidered after major revisions.
Comments on the Quality of English LanguageThroughout the text there are many examples of confused choice of words, grammar mistakes and too complicated syntax, which disrupts the train of thought and readability.
Author Response
Comments 1: This article provides a substantial amount of information in a structured form of descriptive text, figures and tables, but with no introduction being denoted as such. (Introduction as a separate chapter ?) |
Response 1: The situation you mentioned is entirely attributed to our oversight. We have added the necessary heading. We would like to express our deepest apologies for this omission. |
Comments 2: The abstract is also not divided into structural sections. (When not necessary ignore the comment) |
Response 2: We respectfully state that Applied Sciences journal prefers the abstract section not to be divided into structural sections. |
Comments 3: However, in order to be understood more clearly, it needs significant language improvement. Throughout the text there are many examples of confused choice of words, grammar mistakes and too complicated syntax, which disrupts the train of thought and readability. (whole Results chapter!) Response 3: We sincerely apologize for the confusion we have caused, including the grammatical errors and complex syntax in the manuscript. English is not the native language of any of the contributing authors, and as a precaution, we had previously utilized the editing services of the MDPI editing service. If you could kindly specify the particular issues, especially those affecting clarity, we would be more than willing to make any necessary improvements. Additionally, we have attached the editing certificate to the response file. Once again, we deeply apologize for any confusion we may have caused. Comments 4: In the text, the results are often assessed as "good“, "better“, "successful“, or "unsuccessful“ without being referred to the range that defines such classification. (no statistical significance to support statements (numbers)! in rows 23-25, whole Results Chapter! and rows 311-319) |
Response 4: In the course of our study, we aimed to investigate treatment options (endocrown and post-core) applicable in cases of excessive coronal tissue loss in maxillary central incisors with insufficient circumferential ferrule. Our objective was to present a more comprehensive study than previous literature to clinicians. As you may have noticed in your valuable reviews, our study involves numerous groups. Although in the draft of our study, we initially added all numerical values individually to the results section and the result was a continuous and highly complex section with numerous numerical values, making it nearly impossible to comprehend. Furthermore, it is important to note that numerical values obtained from finite element analysis studies are presented as specific values rather than a range. For this reason, we attempted to enhance the clarity of the results section by including the maximum principal stress (Pmax) values for all groups beneath the respective group in numerous figures throughout our study. The aforementioned approach was also one of the recommendations we received from an expert in our field during the draft stage. However, if you, as the esteemed reviewer, still find the results section complex and unclear, and if you have specific recommendations, we express our willingness to make all necessary revisions according to your suggestions. We appreciate your valuable insights and look forward to any guidance you may provide to improve the clarity and understanding of the results section. Thank you for your time and consideration. |
4. Response to Comments on the Quality of English Language |
Point 1: Throughout the text there are many examples of confused choice of words, grammar mistakes and too complicated syntax, which disrupts the train of thought and readability |
Response 1: As previously mentioned, none of the contributing authors to our study are native English speakers. Therefore, the English editing processes of our research paper, which we intend to submit to the Applied Sciences journal under the MDPI umbrella, were conducted by the MDPI editing service. If there are specific sections you would like us to focus on for further revision, we would be more than happy to review and revise them with sincerity. We would like to express our willingness to accommodate any suggestions you may have |
[We express our sincere gratitude for your valuable suggestions and comments. We want to convey that we have endeavored to organize and respond to your valuable remarks with utmost care. We look forward to hearing your further suggestions and wish you well.]

Reviewer 3 Report
Comments and Suggestions for Authors
The subject of the research is a research on a situation that dentists frequently encounter clinically. I think it can be useful for the readers, but some gaps need to be eliminated.
The question of major interest is why FEA analysis was chosen. The question of this research could have been carried out in laboratory conditions under in vitro conditions using a mastication simulator. What prompted the researchers to use FEA analysis? In order to explain this, it would be appropriate to give information about the place and use of FEA analysis in dentistry in the Introduction section. This should also be discussed in a paragraph in the Discussion section.
Abstract: Please rephrase the last sentence (conclusion sentence) since its not clear for the readers.
Introduction: Please correct to “This study aimed to evaluate…”
M&Ms: - Present the groups in paragraphs, that is, by adding indents. It will be more understandable. - In addition, in the abstract section, it is stated that 150N oblique force is applied, while in the M&Ms section it is 100N and the direction is unclear. This confusion should be corrected.
Results: - In all figures please add abbreviations to the legends section. Also if its possible please work on the presentation of the figures since they are not easy to follow. – Figure 8-9 add title to the Y axis. And please correct the values which is converted by Microsoft Excel to 1E+X. Instead you may use superscript numbers by 10.
Discussion: As I said before, the reason for FEA and its limitations need to be discussed. Study limitations should also be clearly discussed at the end of this section.
Coclusion: Regarding point 2, it should be more specific. According to which method it is not an alternative, it should be supportable by the findings in this study. It is too vague as it is.
Sincerely,
Comments on the Quality of English LanguageMinor editing of English language required
Author Response
Comments 1: The question of major interest is why FEA analysis was chosen. The question of this research could have been carried out in laboratory conditions under in vitro conditions using a mastication simulator. What prompted the researchers to use FEA analysis? In order to explain this, it would be appropriate to give information about the place and use of FEA analysis in dentistry in the Introduction section. This should also be discussed in a paragraph in the Discussion section. |
Response 1: Finite Element Analysis (FEA) is commonly employed in the evaluation of complex structures such as dental restorations. In designing this study, we aimed to conduct a comprehensive investigation, leading us to delineate numerous subgroups. Recognizing the need for both model standardization and a judicious consideration of the potential excess in the number of samples, we concluded that the application of FEA would be notably more appropriate under these circumstances. Furthermore, in the discussion section (lines 335-341), we expounded upon the significance of FEA based on valuable insights derived from your recommendations. |
Comments 2: Please rephrase the last sentence (conclusion sentence) since its not clear for the readers. |
Response 2: We appreciate your attention to this matter. Necessary adjustments have been made in lines 28-29. Comments 3: Introduction: Please correct to “This study aimed to evaluate… Response 3: Thank you for your correction. We have made the adjustments as per your request in line 74. |
Comments 4: Present the groups in paragraphs, that is, by adding indents. It will be more understandable Response 4: We express our gratitude for bringing this matter to our attention. The descriptions of the groups have been appended to the end of the figure legends in Fig. 1. We deemed it unnecessary to reiterate the process of group formation, as it has been elaborated in lines 111-113. We hope these revisions meet your expectations. Comments 5: it is stated that 150N oblique force is applied, while in the M&Ms section it is 100N and the direction is unclear. This confusion should be corrected. |
Response 5: We express our infinite gratitude for bringing this matter to our attention. We have rectified the existing errors in lines 127-128. |
Comments 6: In all figures please add abbreviations to the legends section. |
Response 6: All abbreviations and their corresponding expansions have been individually included beneath each figure. We thank you for bringing this to our attention. |
Comments 7: Also if its possible please work on the presentation of the figures since they are not easy to follow. |
Response 7: It's indeed noted that the figures in this study possess high resolution; however, their resolution significantly decreases when converted to PDF format. I plan to address this matter with the journal editor and propose the individual uploading of figures. Additionally, I will upload the current revised file in Word format. I hope to provide you with the high-resolution versions accordingly. |
Comments 8: Figure 8-9 add title to the Y axis. Response 8: We apologize for the oversight. Titles for the Y-axis have been added to Figures 8-9. We appreciate your attention and comment. Comments 9: And please correct the values which is converted by Microsoft Excel to 1E+X. Instead you may use superscript numbers by 10. |
Response 9: The values converted with Microsoft Excel have been corrected. We hope the revisions are now comprehensible. |
Comments 10: As I said before, the reason for FEA and its limitations need to be discussed. Study limitations should also be clearly discussed at the end of this section. Response 10: We appreciate your valuable feedback. In line with your insightful comments, Finite Element Analysis (FEA) has been further discussed in lines 335-341. The limitations of the study were addressed in lines 349-352, shedding light on a crucial aspect that adds depth to our work. Thank you for bringing this important observation to our attention. Comments 11: Regarding point 2, it should be more specific. According to which method it is not an alternative, it should be supportable by the findings in this study. It is too vague as it is. |
Response 11: We appreciate your detailed clarification. As depicted in Figure 9, the Endocrown group, specifically the Endocrown with Circumferential Ferrule (ECER) subgroup, demonstrated the most favorable results in terms of restoration material fatigue performance. However, it also exhibited a less successful outcome for dentin tissue. In our discussion spanning lines 310-328, we delved into the importance and reasons behind this phenomenon. Additionally, despite noting the relatively unsuccessful outcomes for the restoration material in the Endocrown with Inner Retention Form (ECIR) subgroup (as shown in Fig. 9), we reported relatively successful results for dentin tissue and discussed this in lines 329-334. Nevertheless, as stated in the first point of the Conclusion section, we concluded that successful outcomes can be achieved when at least two intact walls and a ferrule effect are established. The consensus among us, the contributing authors, was that the treatment of endodontically treated teeth with excessive coronal destruction involves complex restorations with numerous components. Moreover, we concluded that in cases where a restoration failure may occur, it is more preferable than a potential failure in dentin tissue. Your kind feedback is appreciated, and we hope our response adequately addresses your valuable comments However, should you have any additional feedback or specific requests, please do not hesitate to share. We would be more than pleased to address any further recommendations you may have. We extend our utmost respect to you. |
4. Response to Comments on the Quality of English Language |
Point 1: Minor editing of English language required |
Response 1: With all due respect, we would like to emphasize that none of the contributing authors to our study are native English speakers. Therefore, the English editing processes of our research paper, which we intend to submit to the Applied Sciences journal under the MDPI, were conducted by the MDPI editing service. If there are specific sections you would like us to focus on for further revision, we would be more than happy to review and revise them with sincerity. We would like to express our willingness to accommodate any suggestions you may have |
[We extend our sincere gratitude for your kind remarks at the beginning and your heartfelt wishes at the end. We appreciate your valuable suggestions, and we hope that we have been able to incorporate the necessary revisions as per your insightful comments. Wishing you well.]

Reviewer 4 Report
Comments and Suggestions for Authors
1. Table 1 is incomplete. The table is placed in 2 different pages and the column orientation is disturbed in page 2. Make necessary changes. Some values are missing from raw 3 onwards in table 1. There are columns for compressive strength and shear strength of materials in the table 1, but the value is not available for any material.
2. What does X,Y and in column 2, 3 and 4 in table 2 indicate?
3. The legend box for stress distribution is not visible in figures 2,3,4,5,6, and 7 to analyse the logical deductions of the author.
4. The quality of the images are not sufficient to draw any conclusion. Need better images
5. The mesh convergence study has to added to the work.
6. What does the abbreviation LD in table 3 stands for?
7. What could be the reason for contradictory behaviour in Pmax between resin composite core group and fiber post group as mentioned in line no 162?
8. In line 243, you mentioned that a visco-hyperelastic structure of PDL is incorporated in the study. Can you give an image of the same?
9. What is VS material mentioned in line 253?
10. The fatigue performance for the restoration with LS (figure 8) ranges from 3498*E20 to 2192*E55 cycles. How did you arrive at this value? Do you this this value is realistic?
Comments on the Quality of English LanguageThe Quality of English is poor.
Author Response
Comments 1: Table 1 is incomplete. The table is placed in 2 different pages and the column orientation is disturbed in page 2. Make necessary changes. Some values are missing from raw 3 onwards in table 1. There are columns for compressive strength and shear strength of materials in the table 1, but the value is not available for any material. |
Response 1: We sincerely appreciate your valuable feedback. We prepared the draft of this study using the template file provided by MDPI. However, the mentioned template file disrupts the table placement when page breaks occur. After our adjustments, we managed to fit everything onto a single page. If you could kindly review it again and provide your esteemed insights, we would be more than happy to make any further revisions as per your suggestions. |
Comments 2: What does X,Y and in column 2, 3 and 4 in table 2 indicate? |
Response 2: The phrases mentioned 'X, Y, and Z' describe the axes in the Cartesian coordinate system. However, we deeply apologize for the complexity arising from not specifying this information in the table footnote. The relevant information has been added below Table 2. Comments 3: The legend box for stress distribution is not visible in figures 2,3,4,5,6, and 7 to analyse the logical deductions of the author. Response 3: We would like to extend our deepest apologies for an oversight in the preparation of the figures for this study. The figures, as originally generated, consist of data files of approximately 60 MB each, and when zoomed in their original state, every numeric value is clearly visible. However, during the conversion to PDF format, there has been a significant loss of resolution. We sincerely apologize for this oversight. The revised manuscript, including both Word and PDF formats, will be uploaded. We hope that these revised images will adequately convey the necessary clarity to you. We appreciate your understanding and consideration in this matter. |
Comments 4: The quality of the images are not sufficient to draw any conclusion. Need better images Response 4: As mentioned in our previous response, we firmly believe that we can provide you with figures of better resolution. We sincerely apologize for this deficiency and assure you that, if we are unable to transmit figures with improved resolution, we will contact the editor and request to upload the figures individually. Our heartfelt apologies for any inconvenience Comments 5: The mesh convergence study has to added to the work |
Response 5: Mesh convergence was applied according to the study by Gonder et al., as referenced in line 128. |
Comments 6: What does the abbreviation LD in table 3 stands for? |
Response 6: We deeply regret this omission and extend our sincere respect and gratitude for your thorough review. The requested revision has been made in Table 3 |
Comments 7: What could be the reason for contradictory behaviour in Pmax between resin composite core group and fiber post group as mentioned in line no 162?. |
Response 7: We attempted to address the mentioned issue by discussing it in lines 267-269, referring to the study by Biacchi and Basting. If you find our clarification insufficient or have further insights, we would be pleased to incorporate any additional information you may suggest. |
Comments 8: In line 243, you mentioned that a visco-hyperelastic structure of PDL is incorporated in the study. Can you give an image of the same? Response 8: In fact, we included individual images of the periodontal ligament in Fig. 7. Although simulating its viscoelastic nature through finite element analysis is challenging, an attempt has been made by incorporating Poisson's ratio and Young's Modulus. Comments 9: What is VS material mentioned in line 253? |
Response 9: Once again, we apologize for the oversight and express our gratitude for your thorough review. The requested revision has been made in line 285. |
Comments 10: The fatigue performance for the restoration with LS (figure 8) ranges from 3498*E20 to 2192*E55 cycles. How did you arrive at this value? Do you this this value is realistic? Response 10: We discussed in the section titled 'Calculation of fatigue performance' from lines 135-148 how we obtained the estimated final fracture cycles through cycles applied in a study conducted with Finite Element Analysis (FEA). We believe that without validating this study in a laboratory environment using the mentioned materials, any interpretation of the results lacks a background to assess their realism. However, we understand the basis of your question, and in terms of cycle count, we consider it to be realistic (as mentioned in the limitations of the study in the last paragraph of the Conclusion section, we have eliminated possible human errors and material deficiencies that could occur with finite element analysis). |
4. Response to Comments on the Quality of English Language |
Point 1: The Quality of English is poor. |
Response 1: With all due respect, we would like to emphasize that none of the contributing authors to our study are native English speakers. Therefore, the English editing processes of our research paper, which we intend to submit to the Applied Sciences journal under the MDPI, were conducted by the MDPI editing service. If there are specific sections you would like us to focus on for further revision, we would be more than happy to review and revise them with sincerity. We would like to express our willingness to accommodate any suggestions you may have |
[We extend our sincere gratitude for your kind remarks at the beginning and your heartfelt wishes at the end. We appreciate your valuable suggestions, and we hope that we have been able to incorporate the necessary revisions as per your insightful comments. Wishing you well.]

Round 2
Reviewer 2 Report
Comments and Suggestions for Authors
The article can be accepted for publication after revisions being made.
Reviewer 3 Report
Comments and Suggestions for Authors
Revisions are satisfactory.